# Energy Efficiency-Oriented Resource Allocation for Massive MIMO Systems with Separated Channel Estimation and Feedback

**Feng Hu, Kaiyue Wang, Shufeng Li and Libiao Jin ***

Department of State Key Laboratory of Media Convergence and Communication, School of Information and Communication Engineering, Communication University of China, Beijing 100024, China; fenghu@cuc.edu.cn (F.H.); wangkaiyue_95@163.com (K.W.); shufeng_2004@163.com (S.L.)
* Correspondence: libiao@cuc.edu.cn; Tel.: +86-1880-012-3502

**Abstract:** This paper proposes a dynamic resource allocation scheme to maximize the energy efficiency (EE) for Massive MIMO Systems. The imperfect channel estimation (CE) and feedback are explicitly considered in the EE maximization problem, which aim to optimize the power allocation, the antenna subset selection for transmission, and the pilot assignment. Assuming CE error to be bounded for the complex-constrained Cramer–Rao Bound (CRB), theoretical results show that the lower bound is directly proportional to its number of unconstrained parameters. Utilizing this perspective, a separated and bi-directional estimation is developed to achieve both low CRB and low complexity by exploiting channel and noise spatial separation. Exploiting global optimization procedure, the optimal resource allocation can be transformed into a standard convex optimization problem. This allows us to derive an efficient iterative algorithm for obtaining the optimal solution. Numerical results are provided to demonstrate that the outperformance of the proposed algorithms are superior to existing schemes.

**Keywords:** energy efficiency; massive MIMO; channel estimation; channel capacity; resource allocation

## 1. Introduction

The foreseen demand increasing in data rate has triggered a research race for discovering new ways to enhance the spectral efficiency of the next generation of mobile and wireless networks [1]. Benefiting from spatial multiplexing, massive MIMO systems can enjoy asymptotically orthogonal channels, arbitrary small transmit power, and negligible noise, thus providing significant performance gains in terms of spectral efficiency (SE), security, and reliability compared with conventional MIMO [2]. Furthermore, all of these benefits can be achieved through linear processing with low complexity. However, the usage of a large number of base station (BS) antennas in massive MIMO can significantly increase the radio frequency (RF) circuits and digital signal processing (DSP) power consumption, which has a severe impact on EE [3]. Because of the rapidly rising energy costs and the tremendous carbon footprints of existing systems, EE is gradually accepted as an important design criterion for future communication systems [4,5]. Consequently, research interests are steered to adopt energy-aware network architectures and adjust their operational parameters to optimize the EE performance [6,7].

Nevertheless, most existing works optimizing the resource allocation strategies generally rely on a common assumption that the whole channel characteristics are perfectly known at both the receiver and the transmitter. However, these assumptions seem impractical, especially in frequency division duplexing (FDD) system since noise interference poses significant challenges to channel estimation and channel state information (CSI) feedback. In a time division duplexing (TDD) based massive MIMO system, the CSI in the uplink can be more easily acquired at the BS due to the limited number of users, and then channel reciprocity property can be exploited to realize CSI in downlink via reconstructing

the uplink CSI [8]. However, the CSI acquired in the uplink may be always inaccurate corresponding to the downlink due to the calibration error of radio frequency chains [9]. Especially in FDD protocol, the downlink and uplink CE are necessary, since the channel reciprocity does not hold [10]. There are non-ignorable CE errors under actual transmission conditions, which will significantly affect the SE/EE loss [11].

This work considers the problems of finding the errors of CSI estimation and feedback subject to pilot distortion, ranging from the downlink to uplink, utilizing the theory of a complex-constrained Cramer–Rao bound (CRB) reported in [12] to exactly quantify CE error over a training-based CE technique. The design of separated and bi-directional estimation (SCE) significantly increase the accuracy of CSI feedback. The proposed algorithm is based on singular value decomposition (SVD) of channel matrix as $\mathbf{H} = \mathbf{U}\boldsymbol{\Sigma}\mathbf{V}^{\mathrm{H}}$, where $\mathbf{U}$ and $\mathbf{V}$ are the unitary matrixes, and $\boldsymbol{\Sigma}$ is the diagonal matrix, respectively. $\mathbf{U}$ and $\boldsymbol{\Sigma}$ can be estimated using only received data at both sides by exploiting channel and noise spatial separation. Utilizing CRB, theoretical results show that the lower bound of CE error is directly proportional to its number of unconstrained parameters. Then, the Orthogonal Procrustes (OP) estimating of only the $\mathbf{V}^{\mathrm{H}}$ matrix is more effective than estimating $\mathbf{H}$ directly from the pilot's data.

In practice, the total power consumption of the BS contains not only the transmitting power of the power amplifier (PA) but also power consumption caused by circuit dissipation and signal processing. Meanwhile, activating more transmit antennas enables a higher diversity gain, while the corresponding RF chains consume more circuit and signal processing power [13]. Unlike the existing power allocation schemes that maximize the throughput, the studied scheme maximizes EE by allocating both transmitting power of each sub-channel in reconstructed MIMO architecture and antenna subset selection for transmission, according to the improved CSI feedback and the circuit power consumption. Specifically, the EE optimization can be proven as a standard convex optimization problem. This allows us to derive an efficient iterative algorithm for obtaining the maximum EE boundary. To overcome the shortcomings of the CSI feedback and EE model, the main contributions of the paper are summarized as follows:

- The contributions of proposed SCE are twofold: (1) Directly eliminating the distortion problems to resource allocation, and (2) correctly utilizing the partial CSI from CE to reconstruct MIMO architecture.
- The maximum EE resource allocation strategies with SCE and feedback: The global EE optimization scheme is approximated into a deterministic convex form, and the optimal solutions for the scheme are derived by the quasi-Newton iteration method. Afterwards, the impact of channel estimation error on the EE optimization is developed to highlight the performance improvement of SCE and the feedback model.

The rest of this paper is organized as follows. Section 2 introduces the massive MIMO system, the capacity formulas. Traditional channel estimation (TCE) and feedback is described in Section 3. In Section 4, the separated and bi-directional channel estimation (SCE) and feedback is proposed. Section 5 describes the maximum EE resource allocation strategies with separated CE and feedback. Section 6 presents the simulation results. Finally, conclusions are drawn in Section 7.

Notation: Scalar variables are denoted by normal-face letters, while boldface letters denote vectors and matrices; for a given matrix $\mathbf{A}$, superscripts $\mathbf{A}^{\mathrm{T}}$, $\mathbf{A}^{\mathrm{H}}$ and $\|\mathbf{A}\|_F^2$ represent transpose, conjugate transpose, and the Frobenius matrix norm, respectively; $\mathrm{E}(\cdot)$ denotes the expectation; Notation $\mathrm{tr}(\mathbf{A})$ denotes the trace operator of matrix, and $\mathrm{vec}(\mathbf{A})$ is the vectorization operation of vector $\mathbf{A}$. Operation $\max(a, b)$ denotes returning the maximum element between $a$ and $b$. Operation $\min(a, b)$ denotes returning the minimum element between $a$ and $b$. $\mathrm{Re}(\cdot)$ is the real part operator.

## 2. Massive MIMO Systems Model

In this section, we briefly introduce the massive MIMO systems, and then illustrate the capacity formulas with imperfect CSI.

*2.1. Massive MIMO System*

Consider the downlink of a massive MIMO system for which the transmitter and receiver are equipped with $M$ and $N$ antennas, respectively. The received signal can be modeled by:

$$\mathbf{y}(k) = \sqrt{\frac{P}{M}}\mathbf{H} \cdot \mathbf{x}(k) + \mathbf{z}(k) \tag{1}$$

where $\mathbf{H}$ is the $N \times M$ Rayleigh fading channel transfer matrix whose elements are independent and identically distributed (i.i.d.) complex white Gaussian random variables and $h_{n,m}$ denotes the channel between transmitter antenna $m$ and receiver antenna $n$. $\mathbf{x}$ is the $M \times 1$ transmit symbols and follows a complex normal distribution as $\mathcal{CN}(0, \mathbf{I})$. For $\sqrt{P/M}\mathbf{x}$, the transmission power of each antenna is $P/M$. $\mathbf{z}$ is the $N$-dimensional complex white Gaussian noise with zero mean and variance $\sigma_n^2$.

*2.2. Channel Capacity Analysis*

This section investigates the achievable Channel Capacity (CC) of massive MIMO systems and derives an approximated upper bound on the water filling (WF) for reconstructed architecture with perfect CSI.

2.2.1. Equal Power Distribution with CSI Is Unknown

When the CSI is unknown in the transmitter, the equal power distribution (ED) method is implemented among the transmit antennas. The CC can be written as:

$$C_{\text{ED}} = \log \det \left[ \mathbf{I}_N + \frac{P}{M\sigma_n^2} \cdot \mathbf{HH}^{\text{H}} \right] \tag{2}$$

In practice, the capacity gain obtained by ED is not optimal.

2.2.2. WF with CSI Is Known Perfectly

Assume that the CSI is known perfectly and instantaneously at the transmitter. This assumption is widely adopted for the precoding design problem of massive MIMO system, such that the CSI can be obtained at the receiver via training sequence and subsequently share with the transmitter via limited feedback.

The following derivations are now presented, which shows an approximated upper bound on the achievable CC.

Using SVD to dispose: $\mathbf{H} = \mathbf{U\Sigma V}^{\text{H}}$, where $\mathbf{U}$ and $\mathbf{V}$ are the $N \times N$ and $M \times M$ unitary matrix, respectively. Similarly, eigenvalue decomposition (EVD) of $\mathbf{HH}^{\text{H}}$ can be established: $\mathbf{HH}^{\text{H}} = \mathbf{U\Sigma\Sigma}^{\text{H}}\mathbf{U}^{\text{H}} = \mathbf{U\Lambda U}^{\text{H}}$, where $\mathbf{\Lambda}$ is the diagonal matrix whose diagonal elements can be written as:

$$\lambda_i = \left\{ \begin{array}{l} \sigma_i^2, i = 1, 2, \ldots, r \\ 0, i = r+1, \ldots, N \end{array} \right. \tag{3}$$

where $r$ is the rank of the $\mathbf{HH}^{\text{H}}$. $\sigma_i$ is the square of the singular value of $\mathbf{H}$.

When perfect CSI is available, the precoding by $\mathbf{V}$ at the transmitter and the combination by $\mathbf{U}^{\text{H}}$ at the receiver are implemented. Subsequently, the received signal can be rewritten as:

$$\begin{aligned} \tilde{\mathbf{y}} &= \sqrt{\frac{P}{M}}\mathbf{U}^{\text{H}}\mathbf{HV}\tilde{\mathbf{x}} + \tilde{\mathbf{z}} \\ &= \sqrt{\frac{P}{M}}\mathbf{\Sigma}\tilde{\mathbf{x}} + \tilde{\mathbf{z}} \end{aligned} \tag{4}$$

where $\tilde{\mathbf{z}} = \mathbf{U}^{\text{H}}\mathbf{z}$ which satisfies: $\tilde{\mathbf{z}} \sim \mathcal{CN}(0, \sigma_n^2\mathbf{I})$, $\mathbf{x} = \mathbf{V}\tilde{\mathbf{x}}$ follows a complex normal distribution as $\mathcal{CN}(0, \mathbf{I})$.

Furthermore, $\tilde{\mathbf{y}}$ can be disassembled to $r$ virtual sub-channels: $\tilde{y}_i = \sqrt{\frac{P}{M}}\sqrt{\lambda_i}\tilde{x}_i + \tilde{z}_i, i = 1, 2, \ldots, r$.

Classically, the WF algorithm is popularized to provide a high gain of CC via power allocation in the reconstructed and simplified architecture. In WF, more power is allocated to the sub-channel with higher $\lambda_i$ to maximize the CC of all the sub-channels [14].

By solving the following power distribution problem, the maximum CC can be described in more detail:

$$
\begin{aligned}
C_{\text{WF}} &= \max_{\{\gamma_i\}} \sum_{i=1}^{r} \log_2(1 + \frac{P\gamma_i}{M\sigma_n^2}\lambda_i) \\
&s.t. \sum_{i=1}^{r} \gamma_i = M
\end{aligned}
\tag{5}
$$

where $\gamma_i$ is the transmit power on the $i-th$ sub-channel and satisfies $\sum_{i=1}^{r} \gamma_i = M$ to keep the total power constant.

Using the Lagrange method, the optimal energy distribution is given by:

$$
\gamma_i^{opt} = \max\{(\mu - \frac{M\sigma_n^2}{P\lambda_i}), 0\}
\tag{6}
$$

where $\mu = \frac{M}{r-j+1}[1 + \frac{\sigma_n^2}{P}\sum_{i=1}^{r-j+1}\frac{1}{\lambda_i}]$ is the dynamic threshold decision for activated sub-channel, and $j$ is the iteration times of the WF algorithm [15].

## 3. Traditional CE (TCE) and Feedback

### 3.1. CSI Error Analysis

Practically, the receiver can estimate the channel, based on the observation of the training sequence, as $\hat{\mathbf{H}}$ and the error $\Delta\mathbf{H}$ in estimation is given: $\mathbf{H} = \hat{\mathbf{H}} + \Delta\mathbf{H}$. By well-known properties of the conditional mean, $\mathbf{H}$ and $\hat{\mathbf{H}}$ are uncorrelated identically distributed (i.i.d.) complex white Gaussian random variables. $\Delta\mathbf{H} \sim \mathcal{CN}(0, \sigma_e^2\mathbf{I})$ is a statistical estimated error which is independent of $\hat{\mathbf{H}}$ [9].

Considering the CE error, the receiving signal can be written as:

$$
\begin{aligned}
\tilde{\mathbf{y}} &= \sqrt{\frac{P}{M}}\hat{\mathbf{U}}^H\mathbf{H}\hat{\mathbf{V}}\tilde{\mathbf{x}} + \hat{\mathbf{U}}^H\tilde{\mathbf{z}} \\
&= \sqrt{\frac{P}{M}}\hat{\mathbf{U}}^H\hat{\mathbf{U}}\hat{\mathbf{\Sigma}}\hat{\mathbf{V}}^H\hat{\mathbf{V}}\tilde{\mathbf{x}} + \sqrt{\frac{P}{M}}\hat{\mathbf{U}}^H\Delta\mathbf{H}\hat{\mathbf{V}}\tilde{\mathbf{x}} + \hat{\mathbf{U}}^H\tilde{\mathbf{z}} \\
&= \sqrt{\frac{P}{M}}\hat{\mathbf{\Sigma}}\tilde{\mathbf{x}} + \underbrace{\sqrt{\frac{P}{M}}\hat{\mathbf{U}}^H\Delta\mathbf{H}\hat{\mathbf{V}}\tilde{\mathbf{x}} + \hat{\mathbf{U}}^H\tilde{\mathbf{z}}}_{\mathbf{Z}}
\end{aligned}
\tag{7}
$$

where $\mathbf{Z}$ is the equivalent observation noise, which combines the additive noise and the CE error. CE error increases the total noise power, which can be equivalent to the loss of Signal-to-Noise Ratio (SNR).

The following theorem will derive the approximated variance of equivalent noise.

Let $\mathbf{f} = \sqrt{\frac{P}{M}}\hat{\mathbf{U}}^H\Delta\mathbf{H}\hat{\mathbf{V}}\tilde{\mathbf{x}}$, $\mathbf{G} = \hat{\mathbf{U}}^H\Delta\mathbf{H}\hat{\mathbf{V}}$; it can be obtained from the expression of $\mathbf{G}$:

$$
\begin{aligned}
\text{E}(\mathbf{G}\mathbf{G}^H) &= \text{E}(\hat{\mathbf{U}}^H\Delta\mathbf{H}\hat{\mathbf{V}}\hat{\mathbf{V}}^H\Delta\mathbf{H}^H\hat{\mathbf{U}}) \\
&= \mathbf{U}^H\text{E}(\Delta\mathbf{H}\Delta\mathbf{H}^H)\mathbf{U} \\
&= \sigma_e^2\mathbf{I}_{N\times N}
\end{aligned}
\tag{8}
$$

Hence, **G** and $\Delta\mathbf{H}$ have the same probability distribution. The channel noise in **Z** can be described as follows:

$$
\begin{aligned}
\mathbf{f} &= \sqrt{\frac{P}{M}}
\begin{bmatrix}
g_{11} & \cdots & g_{1M} \\
\vdots & \ddots & \vdots \\
g_{N1} & \cdots & g_{NM}
\end{bmatrix}
\begin{bmatrix}
\sqrt{\gamma_1} & 0 & \cdots & 0 \\
0 & \sqrt{\gamma_2} & \cdots & 0 \\
\vdots & \vdots & \ddots & \vdots \\
0 & 0 & \cdots & \sqrt{\gamma_M}
\end{bmatrix}
\begin{bmatrix}
x_1 \\
x_2 \\
\vdots \\
x_M
\end{bmatrix} \\
&= \sqrt{\frac{P}{M}}
\begin{pmatrix}
\sum\limits_{i=1}^{M} g_{1i} \cdot \sqrt{\gamma_i} \cdot x_i \\
\sum\limits_{i=1}^{M} g_{2i} \cdot \sqrt{\gamma_i} \cdot x_i \\
\vdots \\
\sum\limits_{i=1}^{M} g_{Ni} \cdot \sqrt{\gamma_i} \cdot x_i
\end{pmatrix}
\end{aligned}
\tag{9}
$$

The variance of $f_d = \sum\limits_{i=1}^{M} g_{di} \cdot \sqrt{\gamma_i} \cdot x_i, d = 1, 2, \ldots, N$, can be written as:

$$
\begin{aligned}
& \mathrm{E}\left\{ [f_d - \mathrm{E}\{f_d\}] \cdot [f_d - \mathrm{E}\{f_d\}]^* \right\} \\
&= \mathrm{E}\left\{ \left[ \sqrt{\frac{P}{M}} \sum_{i=1}^{M} g_{di} \cdot \sqrt{\gamma_i} \cdot x_i \right] \cdot \left[ \sqrt{\frac{P}{M}} \sum_{i=1}^{M} g_{di} \cdot \sqrt{\gamma_i} \cdot x_i \right]^* \right\} \\
&= \frac{P}{M} \sum_{i=1}^{M} \gamma_i \mathrm{E}\left\{ |g_{di}|^2 \right\} \cdot \mathrm{E}\left\{ |x_i|^2 \right\} \\
&= P\sigma_e^2
\end{aligned}
\tag{10}
$$

In conclusion, the equivalent noise **Z** follows a complex normal distribution as:

$$
\begin{aligned}
\mathbf{Z} &\sim \mathcal{CN}(0, \sigma_{\hat{n}}^2 \mathbf{I}) \\
\sigma_{\hat{n}}^2 &= P\sigma_e^2 + \sigma_n^2
\end{aligned}
\tag{11}
$$

### 3.2. TCE in Receiver and Uplink Feedback

In Figure 1, the CSI based power allocation procedure is described in detail. Based on characteristics of the pilot sequences, the estimated $\hat{\mathbf{H}}$ is obtained by CE in the downlink, and then shared with the transmitter via feedback in the uplink. However, except for CE error, there is distortion in the uplink, which ulteriorly deteriorates the accuracy of feedback CSI.

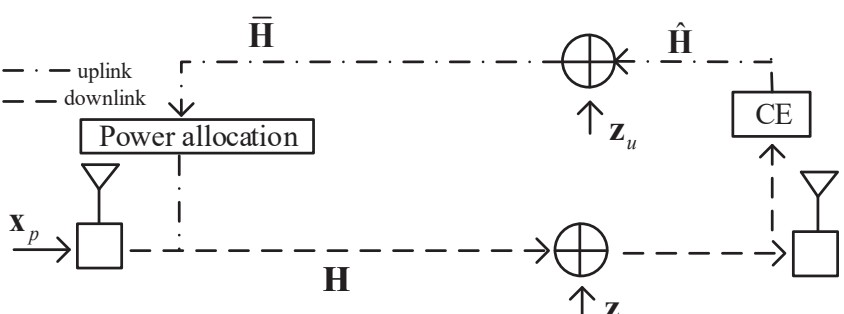

**Figure 1.** Traditional CE and feedback model.

The received pilot sequence in the downlink can be modeled as:

$$\mathbf{y}_p = \sqrt{\frac{P}{M}} \mathbf{H} \mathbf{x}_p + \mathbf{z} \tag{12}$$

Assume the channel has been used for a total of $Q$ symbol transmissions. Among all these transmissions, the initial $L$ symbols are known as training symbols and the observed outputs are thus training outputs. Stacking the training symbols as a matrix, we have $\mathbf{x}_p = [\mathbf{x}_1, \mathbf{x}_2, \ldots, \mathbf{x}_L]$, where the pilot sequence is orthogonal.

$\hat{\mathbf{H}}$ can be estimated by using the observation of the training sequence in the receiver. Then, a general result is presented to quantify the improvement in estimation accuracy of traditional CE and and the feedback model.

To simplify the EE optimization, we only make a detailed derivation of the CE error in $\hat{\mathbf{H}}$ and the CRB bound of the CE scheme. The downlink interference terms corresponding to CE error and channel noise have been approximated to $\mathbf{Z} \sim \mathcal{CN}(0, \sigma_{\hat{h}}^2 \mathbf{I})$ in Section 3.1. We proceed to derive the SNR loss for uplink feedback. The effect of uplink feedback can be modeled as an unfaded AWGN channel [16–18].

**Proof.** The details of the proof for the achieved uplink channel model can be found in Appendix A.　□

### 3.3. CE Error Bound for $\hat{\mathbf{H}}$

This paper utilizes the theory of CRB reported in [12] to exactly quantify CE error over a training-based CE technique. The lower bound of the CE error is proportional to the number of unconstrained real parameters needed to describe $\mathbf{H}$:

$$\mathrm{E}(\|\mathbf{H} - \hat{\mathbf{H}}\|_{\mathrm{F}}^2) \geq \Psi \frac{\sigma_{\mathrm{n}}^2}{2\sigma_{\hat{s}}^2 L} \tag{13}$$

For the TCE method, the error bound for estimation of matrix $\mathbf{H}$ from the reference data $\mathbf{x}_p$ is given as:

$$\mathrm{E}(\|\mathbf{H} - \hat{\mathbf{H}}\|_{\mathrm{F}}^2) \geq MN \frac{\sigma_{\mathrm{n}}^2}{\sigma_{\hat{s}}^2 L} \tag{14}$$

where $2MN$ is the number of the parameters required to describe the complex $N \times M$ channel matrix $\mathbf{H}$, and $\hat{\mathbf{H}}$ is any estimation of $\mathbf{H}$. It is evident that the lower limit of the CE error $\Delta\mathbf{H}$ exists:

$$\begin{aligned}
\Delta\mathbf{H} &\sim \mathcal{CN}(0, \sigma_e^2 \mathbf{I}) \\
\sigma_e^2 &= \frac{1}{MN} \min \mathrm{E}(\|\mathbf{H} - \hat{\mathbf{H}}\|_{\mathrm{F}}^2)) \\
&= \frac{\sigma_{\mathrm{n}}^2}{\sigma_{\hat{s}}^2 L}
\end{aligned} \tag{15}$$

### 3.4. The Uplink Distortion for Feedback

In previous works, instead of idealizing the feedback channel as a fixed-rate, error-free bit pipe, transmission from each receiver to the transmitter over the noisy feedback channel is considered. Assume that the downlink channel is a faded channel and the dedicated uplink channel can be modeled as an unfaded AWGN channel. In this work, we refer to the channel model [16–18]. The corresponding schematic is illustrated in Figure 1, where the terminals directly feed the estimated CSI (TCE) or received training pilots (SCE) back to the BS. Note that the terminals use analog linear modulation to transmit the feedback data, where the terminals directly modulate the carrier using the transfer form in [19]. On this basis, the feedback CSI for power allocation can be modeled:

$$\bar{\mathbf{H}} = \hat{\mathbf{H}} + \mathbf{z}_u \tag{16}$$

where $\mathbf{z}_u$ is the uplink channel matrix and satisfies: $\mathbf{z}_u \sim \mathcal{CN}(0, \sigma_{n_u}^2 \mathbf{I})$.

Based on Equations (11) and (15), the CE error and the uplink distortion $\mathbf{z}_u$ in Appendix A can be integrated as overlap-added noise. Rearranging Equation (11) via confirming Equation (15) and overlapping uplink distortion in Equation (16), the total equivalent noise $\sigma_t^2$ can be given by:

$$\sigma_t^2 = \sigma_n^2 + (\sigma_{n_u}^2 + \sigma_e^2)P \tag{17}$$

Then, based on Equations (4) and (5), the Signal-to-Interference-plus-Noise Ratio (SINR) on the WF condition experienced at the equivalent noise can be obtained as:

$$\text{SINR} = \frac{P \gamma_i \widehat{\lambda}}{M[\sigma_n^2 + (\sigma_{n_u}^2 + \sigma_e^2)P]} \tag{18}$$

As considered in some previous studies in this field [11], the achieved downlink SINR for the CC algorithm in Equation (5) is closely related to the total equivalent noise. Mathematically stating:

$$C_{\text{TCE−CSI}} = \sum_{i=1}^{r} \log_2 \left(1 + \frac{P \gamma_i}{M[\sigma_n^2 + (\sigma_{n_u}^2 + \sigma_e^2)P]} \widehat{\lambda}_i \right) \tag{19}$$

where $\sigma_t^2 = \sigma_n^2 + (\sigma_{n_u}^2 + \sigma_e^2)P$.

Both the CE error and the uplink distortion will significantly lead to a decrease of CC. Commonly for a certain CSI distortion, the CC performance via power allocation optimizing is even worse than that through ED.

## 4. The Separated and Bi-Directional Channel Estimation (SCE) and Feedback

To overcome the drawbacks of CSI distortion in TCE and feedback, the design of separated and bi-directional estimation can significantly increase the accuracy of CSI feedback. The corresponding schematics is illustrated in Figure 2. Different from Figure 1, the distorted pilot at the receiver is directly fed back to the transmitter without CE procedure in the downlink.

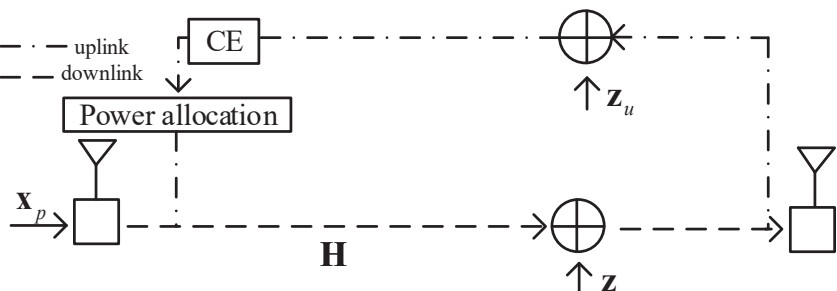

**Figure 2.** The separated CE and feedback model.

Assume that the received pilots $\mathbf{y}_p$ in the downlink is modeled in Equation (12). To simplify the process of feedback and efficiently utilize the spatial characteristic construct in Equation (4), the spatially selective noise filtration is implemented via the feedback relative to transmitter observation. More concretely, the feedback of received pilot in the uplink in Appendix A is formulated as:

$$\begin{aligned} \bar{\mathbf{y}} &= \mathbf{y}_p + \mathbf{z}_u \\ &= \sqrt{\tfrac{P}{M}} \mathbf{H} \mathbf{x}_p + \mathbf{z}' \end{aligned} \tag{20}$$

where $\mathbf{z}'=\mathbf{z}+\mathbf{z}_u \sim \mathcal{CN}(0, \sigma_{n'}^2 \mathbf{I})$, $\sigma_{n'}^2 = \sigma_n^2 + \sigma_{n_u}^2$ represents the AWGN noise variance both in the uplink and downlink.

The proposed separated CE algorithm is based on the SVD process: $\mathbf{H} = \mathbf{U\Sigma V}^{\mathrm{H}}$. $\mathbf{U}$ and $\mathbf{\Sigma}$ can be estimated using only received data at both sides by exploiting channel and noise spatial separation. Utilizing CRB, theoretical results show that that the lower bound of CE error is directly proportional to its number of unconstrained parameters. Then, in the BS, the orthogonal procrustes (OP) estimating only the $\mathbf{V}^{\mathrm{H}}$ matrix can perform more efficiently than estimating $\mathbf{H}$ directly from the pilot's data. The following theorem will derive the approximated estimation of $\mathbf{U}$, $\mathbf{\Sigma}$ and $\mathbf{V}^{\mathrm{H}}$.

*4.1. The Estimation of $\mathbf{U}$ and $\mathbf{\Sigma}$ with Spatially Selective Noise Filtration*

Making the following assumptions for signal separation in Equation (20):

- $\mathbf{x}_P$ are assumed to be spatially and temporally independent with identical source power: $\sigma_s^2 = \frac{P}{M}$:

$$\mathrm{E}\{\mathbf{x}_p(k)\mathbf{x}_p{}^{\mathrm{H}}(l)\} =\delta(k,l)\sigma_s^2 \mathbf{I} = \left\{ \begin{array}{l} \sigma_s^2, k = l \\ \mathbf{0}, k \neq l \end{array} \right. \tag{21}$$

where $k$ and $l$ represent the time instant.
- $\mathbf{z}'$ is spatio-temporally white additive Gaussian noise such that:

$$\mathrm{E}\{\mathbf{z}'(k)\mathbf{z}'^{\mathrm{H}}(l)\} =\delta(k,l)\sigma_{n'}^2 \mathbf{I} = \left\{ \begin{array}{l} \sigma_{n'}^2, k = l \\ \mathbf{0}, k \neq l \end{array} \right. \tag{22}$$

- The source signal $\mathbf{x}_P$ and additive noise $\mathbf{z}'$ are statistically independent:

$$\mathrm{E}\{\mathbf{x}_p(k)\mathbf{z}'^{\mathrm{H}}(k-l)\} =\mathbf{0} \tag{23}$$

Using (21–23), the correlation matrix of $\bar{\mathbf{y}}$ can be described:

$$\begin{aligned} \mathbf{R}_{\bar{y}} &=\mathrm{E}\{\bar{\mathbf{y}}\bar{\mathbf{y}}^{\mathrm{H}}\} \\ &=\mathrm{E}\{(\sqrt{\tfrac{P}{M}}\mathbf{H}\mathbf{x}_p+\mathbf{z}')(\sqrt{\tfrac{P}{M}}\mathbf{x}_p{}^{\mathrm{H}}\mathbf{H}^{\mathrm{H}}+\mathbf{z}'^{\mathrm{H}})\} \\ &=\tfrac{P}{M}\mathbf{H}\mathbf{H}^{\mathrm{H}}+\sigma_{n'}^2\mathbf{I} \end{aligned} \tag{24}$$

where $\sigma_{n'}^2$ denotes the overall noise power both in the downlink and uplink.

Performing EVD on $\mathbf{R}_{\bar{y}}$, (24) can be rewritten as:

$$\mathbf{R}_{\bar{y}} = [\ \mathbf{U}, \quad \mathbf{U}_{\mathrm{n}}] \left[ \begin{array}{cc} \mathbf{\Lambda}_s & 0 \\ 0 & \mathbf{\Lambda}_{\mathrm{n}} \end{array} \right] \left[ \begin{array}{c} \mathbf{U}^{\mathrm{H}} \\ \mathbf{U}_{\mathrm{n}}^{\mathrm{H}} \end{array} \right] \tag{25}$$

$$\mathbf{\Lambda}_s = \mathrm{diag}\left(\frac{P}{M}\lambda_1 + \sigma_{n'}^2, \ldots, \frac{P}{M}\lambda_r + \sigma_{n'}^2\right)_{r\times r} \tag{26}$$

$$\mathbf{\Lambda}_n = \mathrm{diag}\left(\sigma_{n'}^2, \ldots, \sigma_{n'}^2\right)_{(N-r)\times(N-r)} \tag{27}$$

where $\mathbf{\Lambda}_s$ and $\mathbf{U}$ represent the eigenvalues and eigenvectors of the matrix $\mathbf{H}$, and $\mathbf{\Lambda}_{\mathrm{n}}$ and $\mathbf{U}_{\mathrm{n}}$ represent the eigenvalues and eigenvectors of the noise subspace.

Assuming that $\sigma_{n'}^2$ is known exactly, $\mathbf{U}$ and $\mathbf{\Sigma}$ can be obtained on both sides via spatially selective noise filtration from only received data:

$$\mathbf{\Sigma}=\frac{\mathbf{\Lambda}_s - \sigma_{n'}^2\mathbf{I}}{P/M} \tag{28}$$

where $\lambda_i$ can be separated by spatially selective noise filtration. In addition, $\sigma_{n'}^2$ in the downlink and uplink can be eliminated in this case for power allocation.

*4.2. The OP Estimation of $\mathbf{V}^H$*

OP Estimation

　　When $\mathbf{R} = \mathbf{U}\boldsymbol{\Sigma}$ can be accurately estimated, let $\|.\|_F^2$ denote the Frobenius matrix norm. Then, the following optimization problem is given to describe the process of applying $\mathbf{x}_p$ to estimate $\mathbf{V}^H$:

$$\min \left\| \bar{\mathbf{y}} - \sqrt{\frac{P}{M}} \mathbf{R} \mathbf{V}^H \mathbf{x}_p \right\|_F^2 \tag{29}$$

where $(\mathbf{V}^H \mathbf{x}_p)^H \mathbf{V}^H \mathbf{x}_p = \mathbf{I}$. From [20], the optimization problem can be regarded as a generalization of the orthogonal procrustes (OP) problem. The Frobenius matrix norm can be rewritten as:

$$
\begin{aligned}
&\left\| \bar{\mathbf{y}} - \sqrt{\frac{P}{M}} \mathbf{R} \mathbf{V}^H \mathbf{x}_p \right\|_F^2 ) \\
&= \mathrm{tr}(\bar{\mathbf{y}}^H \bar{\mathbf{y}}) + \frac{P}{M} \mathrm{tr}(\mathbf{R}^H \mathbf{R}) - 2\sqrt{\frac{P}{M}} \mathrm{tr}((\mathbf{V}^H \mathbf{x}_p)^H \mathbf{R}^H \bar{\mathbf{y}})
\end{aligned}
\tag{30}
$$

Hence, the minimization of Equation (30) is equivalent to the maximum of the function $\mathrm{tr}((\mathbf{V}^H \mathbf{x}_p)^H \mathbf{R}^H \bar{\mathbf{y}})$. Letting $\mathbf{R}^H \bar{\mathbf{y}} = \mathbf{U}'\boldsymbol{\Sigma}'\mathbf{V}'^H$ be a SVD, and $\mathrm{tr}((\mathbf{V}^H \mathbf{x}_p)^H \mathbf{R}^H \bar{\mathbf{y}})$ can be rewritten as:

$$
\begin{aligned}
&\mathrm{tr}((\mathbf{V}^H \mathbf{x}_p)^H \mathbf{R}^H \bar{\mathbf{y}}) \\
&= \mathrm{tr}((\mathbf{V}^H \mathbf{x}_p)^H \mathbf{U}'\boldsymbol{\Sigma}'\mathbf{V}'^H) \\
&= \mathrm{tr}(\mathbf{W}\boldsymbol{\Sigma}') \\
&= \sum_{i=1}^n w_{ii}\rho_i \\
&\leq \sum_{i=1}^n \rho_i
\end{aligned}
\tag{31}
$$

where $\mathbf{W} = \mathbf{V}'^H (\mathbf{V}^H \mathbf{x}_p)^H \mathbf{U}'$. The equation can only be established when $\mathbf{W} = \mathbf{I}$. Hence, the OP solution is given by:

$$\mathbf{V}'^H (\mathbf{V}^H \mathbf{x}_p)^H \mathbf{U}' = \mathbf{I} \Rightarrow \hat{\mathbf{V}}^H = \mathbf{U}'\mathbf{V}'^H \mathbf{x}_p{}^H \tag{32}$$

　　As $\mathbf{R}$ is perfectly known in Section 4.1, the error of the CE is directly determined by the accuracy of $\mathbf{V}^H$. This error is directly caused by the embedded noise in $\mathbf{E_r}$, which is detailed as:

$$
\begin{aligned}
\mathbf{E_r} &= \mathbf{R}^H \bar{\mathbf{y}} \\
&= \mathbf{R}^H (\mathbf{H}\mathbf{x}_p + \mathbf{Z}') \\
&= \underbrace{\mathbf{R}^H \mathbf{R} \mathbf{V}^H \mathbf{x}_p}_{\mathbf{E_{rr}}} + \underbrace{\mathbf{R}^H \mathbf{z}'}_{\mathbf{E_p}}
\end{aligned}
\tag{33}
$$

　　In fact, the actual $\mathbf{V}^H$ can be recovered from $\mathbf{E_{rr}}$ by performing SVD. $\mathbf{E_p}$ is the root cause of the $\mathbf{V}^H$ estimation error. Let $\mathbf{e} = \mathbf{E_r} - \mathbf{E_{rr}}$, the autocorrelation function of $\mathbf{e}$ can be described as:

$$
\begin{aligned}
\mathbf{R}_e &= \mathrm{E}\{\mathbf{e}\mathbf{e}^H\} \\
&= \mathrm{E}\{\mathbf{R}^H \mathbf{z}' (\mathbf{R}^H \mathbf{z}')^H\} \\
&= \sigma_{n'}^2 \mathrm{E}\{\mathbf{R}^H \mathbf{R}\} \\
&= \sigma_{n'}^2 \mathrm{E}\{(\mathbf{U}_1 \boldsymbol{\Delta})^H \mathbf{U}_1 \boldsymbol{\Delta}\} \\
&= \sigma_{n'}^2 \mathrm{E}\{\boldsymbol{\Delta}^H \boldsymbol{\Delta}\}
\end{aligned}
\tag{34}
$$

The MSE of $\mathbf{E}_{\mathbf{rr}}$ is given as:

$$\mathbf{E}_{\mathbf{rr}_{\mathbf{MSE}}} = \| \mathbf{E_r} - \mathbf{E_r} \| = \|\mathbf{e}\|_{\mathrm{F}}^2 = \mathrm{tr}\{\mathbf{R}_e\} = \sigma_{n'}^2 \mathrm{E}\{\sum_i^n \eta_i\} \tag{35}$$

where $\eta_i$ represents the eigenvalue of the channel autocorrelation matrix, and $n = \min(N, M)$. For a fixed SNR, the error of $\mathbf{E_r}$ is determined by the sum of eigenvalues of the channel autocorrelation matrix. It can be proven that a smaller sum of eigenvalues of the channel autocorrelation matrix offers a more accurate $\mathbf{V}^{\mathrm{H}}$ estimation [21].

### 4.3. CE Error for $\hat{\mathbf{H}}$

When $\mathbf{R}$ is exactly known by spatially selective noise filtration, in view of Equation (12), the bound for the error of estimation of $\hat{\mathbf{H}} = \mathbf{R}\hat{\mathbf{V}}^{\mathrm{H}}$ is:

$$\mathrm{E}(\|\mathbf{H} - \hat{\mathbf{H}}\|_{\mathrm{F}}^2) \geq \frac{M^2 \sigma_{\mathrm{n}}^2}{2\sigma_s^2 L} \tag{36}$$

where $M^2$ is the number of parameters required to describe a complex $M \times M$ unitary matrix $\mathbf{V}^{\mathrm{H}}$. It is evident that the lower limit of the CE error $\Delta\mathbf{H}'$ exists:

$$\begin{aligned} \Delta\mathbf{H}' &\sim \mathcal{CN}(0, \sigma_{e'}^2 \mathbf{I}) \\ \sigma_{e'}^2 &= \frac{1}{MN} \min \mathrm{E}(\|\mathbf{H} - \hat{\mathbf{H}}\|_{\mathrm{F}}^2) \\ &= \frac{M\sigma_{n'}^2}{2N\sigma_s^2 L} \end{aligned} \tag{37}$$

It is cleared that: $\sigma_{e'}^2 < \sigma_e^2$ when $N > M$, where $\sigma_{e'}^2$ is the lower limit variance of CE error in a separated CE model.

Based on Equations (11) and (37), the total equivalent noise $\sigma_{t'}^2$ in SCE model can be given by:

$$\sigma_{t'}^2 = \sigma_n^2 + \sigma_{e'}^2 P \tag{38}$$

Considering the improved CE and feedback model, CC algorithm in Equation (5) can be rewritten as:

$$C_{\mathrm{SCE-CSI}} = \sum_{i=1}^r \log_2 (1 + \frac{P\gamma_i}{M(\sigma_n^2 + \sigma_{e'}^2 P)} \lambda_i) \tag{39}$$

To this end, the upper bound on the achievable CC with method is derived. What's more, the exact eigenvalues $\lambda_i$ for power allocation can be obtained.

The two competing techniques (TCE and SCE) mentioned in Sections 3.4 and 4.3 are compared as:

(a) In the following conditions from Equations (21)–(23), $\mathbf{U}$ and $\boldsymbol{\Sigma}$ are exactly known. $\hat{\mathbf{V}}^{\mathrm{H}}$ is any unbiased estimate of $\mathbf{V}^{\mathrm{H}}$. Note that $2MN$ is the number of the parameters required to describe the complex $N \times M$ channel matrix $\mathbf{H}$, while $M^2$ is the number of parameters required to describe a complex $M \times M$ unitary matrix $\mathbf{V}^{\mathrm{H}}$. In particular, as the receiving antennas $N$ increases, the number of the parameters needed to estimate $\mathbf{H}$ increases, while that for $\mathbf{V}^{\mathrm{H}}$ remains constant value $M^2$. This can be expected since, as $N$ increases, the complexity of estimating $\mathbf{H}$ increases while the estimation of $\mathbf{V}^{\mathrm{H}}$ remains the constant value.

(b) Under CRB, the minimum estimation error in a channel matrix is directly proportional to the number of unconstrained real parameters required for description. In fact, obviously, from Equations (14) and (36), one can find that the proposed OP algorithm can achieve about $2N/M$ gain over the estimating $\mathbf{H}$ method in terms of minimum estimation error for the same orthogonal training pilots. The estimation gain significantly increases as the number of receive antennas increases.

(c) When the statistical characteristics of CE error is achieved in CRB boundary conditions, the total equivalent noise for downlink is given in Equations (17) and (38) for TCE and SCE, respectively. It is clear that:

$$
\begin{aligned}
&\sigma_{t'}^2 < \sigma_t^2 \\
&s.t. \sigma_{t'}^2 = \sigma_{e'}^2 P + \sigma_n^2 \\
&\sigma_t^2 = (\sigma_e^2 + \sigma_{n_u}^2)P + \sigma_n^2 \\
&\sigma_{e'}^2 < \sigma_e^2
\end{aligned}
\tag{40}
$$

(d) The separated and bi-directional channel estimation is a distributed and parallel computing strategy, which has great advantages in computational complexity and delay, especially for the receiver.

## 5. The Maximum EE in Separated CE and a Feedback Model

The power consumption will increase when the number of transmit antennas increases in practical systems. Thus, the optimal CC obtained by using the ideal threshold value in WF is overoptimistic, and the corresponding transmit antennas is more than the accurate value. To overcome this shortcoming, we consider the EE optimization problem with the optimal active antenna subset in the reconstructed MIMO systems. Specifically, the number of transmit antennas at the transmitter has a significant impact on the EE. To increase the CC, more transmit antennas are required to be activated to exploit a higher diversity gain. However, allowing more antennas to be active will increase the circuit power consumption at the transmitter [22]. To constrain the circuit power consumption, the achievable EE can be established as:

$$
\eta_{\mathrm{EE}} = \frac{C}{P_{\mathrm{sum}}}
\tag{41}
$$

where $P_{\mathrm{sum}}$ denotes the total power consumption which contains not only the power consumption at the transmitter, but also a transmission independent power representing the power consumed by circuit dissipation. $P_{\mathrm{sum}}$ can be written as:

$$
P_{\mathrm{sum}} = P_{\mathrm{t}} + P_{\mathrm{c}}
\tag{42}
$$

where $P_{\mathrm{t}} = \frac{P}{\eta}$ donates the power consumption at the transmitter, $\eta \in (0, 1)$ is the power amplifier efficiency, and $P_{\mathrm{c}}$ is the total circuit power consumption. $P_{\mathrm{c}} = |\psi| \left(P_{\mathrm{DAC}} + P_{\mathrm{filt}} + P_{\mathrm{mix}}\right) + P_{\mathrm{sta}}$, where $\psi$ denotes the activated RF chains set of the transmitter, and $|\psi| \subseteq \{1, \ldots, M\}$, $|A|$ denotes the size of set $A$. $P_{\mathrm{DAC}}$, $P_{\mathrm{filt}}$, $P_{\mathrm{mix}}$, and $P_{\mathrm{sta}}$ are the power consumption of digital to analog converter, filter, mixer, and the static power, respectively [13].

To analyze the maximal EE with respect to the parameters of a massive MIMO system, the following optimization problem is formulated. By substituting Equation (42) into Equation (41), the optimization problem can be given as:

$$
\max \eta_{\mathrm{EE}} = (1 - \tau) \frac{\sum\limits_{i=1}^{|\psi|} \log_2\left(1 + \frac{P\gamma_i}{M(\sigma_n^2 + \sigma_{e'}^2(\tau)P)}\lambda_i\right)}{\frac{P}{\eta} + |\psi|P_1 + P_{\mathrm{sta}}}
\tag{43}
$$
$$
\text{s.t. : } |\psi| \leq r
$$

where $\lambda_i$ is decreasing in order. $r$ is the number of available RF chains and $r = \min(N, M)$. $\tau = L/Q$ represents the pilot overhead (PO); $\sigma_{e'}^2(\tau)$ is related to the number of pilots based on Equation (13). $P_1 = P_{\mathrm{DAC}} + P_{\mathrm{filt}} + P_{\mathrm{mix}}$. $|\psi|P_1$ denotes the circuit power consumption proportional to the number of the active transmitted antennas. $P_{\mathrm{sta}}$ denotes the static power which is independent of both $P$ and $|\psi| P_1$, including the power consumption of the baseband processing, etc.

As $|\psi|$ increases, the sum of CC increases, but the gains grow slowly. When the CC gain is small for a large $|\psi|$, the circuit power consumption dominates EE. There indeed exists a convex optimal $\psi^*$ to achieve the maximum energy efficiency.

The influence of transmit power on the objective function is considered. Equation (43) can be rewritten as:

$$\eta_{\text{EE}'} = \frac{(1-\tau)\sum\limits_{i=1}^{|\psi^*|} \log_2(1 + \frac{P\gamma_i}{M(\sigma_n^2+\sigma_{n'}^2 M^2/2LN)}\lambda_i)}{\frac{P}{\eta} + |\psi^*|\, P_1 + P_2} = \frac{f(P)}{g(P)} \tag{44}$$

If the objective function is a fractional programming problem and the objective function is pseudo-concave, then any stagnation point is a global maximum point. The following derivation proves that the objective function is pseudo-concave with respect to the transmit power.

Take the first derivative of $f(P)$:

$$f'(P) = \sum_{i=1}^{|\psi^*|} \frac{\frac{(1-\tau)\gamma_i\lambda_i}{M(\sigma_n^2+\sigma_{n'}^2 M^2/2LN)}}{\left(1 + \frac{P\gamma_i\lambda_i(1-\tau)}{M(\sigma_n^2+\sigma_{n'}^2 M^2/2LN)}\right)\ln 2} \tag{45}$$

Take the second derivative of $f(P)$:

$$f''(P) = -\sum_{i=1}^{|\psi^*|} \frac{\left(\frac{(1-\tau)\gamma_i\lambda_i}{M(\sigma_n^2+\sigma_n^2 M^2/2LN)}\right)^2}{\left(1 + \frac{P\gamma_i\lambda_i(1-\tau)}{M(\sigma_n^2+\sigma_n^2 M^2/2LN)}\right)^2 \ln 2} < 0 \tag{46}$$

where $f''(P) < 0$, $f(P)$ is a concave function of the transmit power, and $g(P)$ is a linear function of the transmit power. Therefore, the objective function is pseudo-concave and there exists a unique transmit power to achieve the maximum energy efficiency.

Taking maximum $\eta_{EE}(P, \gamma_i, |\psi|)$ as the objective function, optimization is used to achieve the global optimized parameters in this work. Specifically, for the operational parameters $|\psi| \subseteq \{1, \ldots, M\}$, $P > 0, \gamma_i > 0, i = 1, \ldots, |\psi|$, $\mathbf{q} = [q_1, q_2, \ldots, q_n]$ is as a substitute to the global optimization problem. Then, the quasi-Newton global optimization method is used to reconstruct its parameters to improve search performance and computation speed, as

$$f(\mathbf{q}^{(k)}) = \max \eta_{EE}\left(\mathbf{q}^{(k)}\right) \tag{47}$$

where $\mathbf{q}^{(k)} = [q_1^{(k)}, q_2^{(k)}, .., q_m^{(k)}, .., q_n^{(k)}]$ is assigned to $f(\mathbf{q}^{(k)})$. The Quasi-Newton method defines $\mathbf{d}^{(k)} = -\mathbf{Q}(\mathbf{q}^{(k)})^{-1}\nabla f(\mathbf{q}^{(k)})$ as the next search direction of $\mathbf{q}^{(k+1)}$ in practical operation, where the gradient of a function $f(\mathbf{q}^{(k)})$ is denoted as $\nabla f(\mathbf{q}^{(k)})$ and the Hessian matrix is denoted as $\mathbf{Q}(\mathbf{q}^{(k)}) = \nabla^2 f(\mathbf{q}^{(k)})$. The Quasi-Newton method approximates the objective function Hessian inverse using rank-one (SR1) Algorithm in [23].

## 6. Simulation Results

In this section, numerical simulations are conducted to evaluate the performance of the proposed CE method and EE optimization algorithm. We consider a FDD massive MIMO system with an equal number of transmit and receive antennas. Results are obtained over uncorrelated Rayleigh fading channels in the downlink. The training pilot is extracted from a Gaussian distribution with zero mean and variance equal to $P/M$ at the transmitter. For simplicity, the variance of additive white Gaussian noise is assumed to be fixed with different SNR cases. The main parameters for the simulation are provided in Table 1.

**Table 1.** Simulation parameters.

| Parameters | Value |
| --- | --- |
| Efficiency of power amplifier $\eta$ | 0.35 |
| Digital-to-analog converter power $P_{\text{DAC}}$ | 15 mW |
| Mixer power $P_{\text{mix}}$ | 30 mW |
| Filter power $P_{\text{filt}}$ | 3 mW |
| Static power $P_{\text{sta}}$ | 160 mW |
| Noise power in the downlink $\sigma_n^2$ | 10 dBm |
| Noise power in the uplink $\sigma_{n_u}^2$ | 6 dBm |
| PO $\tau$ | 0.05, 0.1 |
| Number of transmit antennas $M$ | 32 |
| Number of receive antennas $N$ | 32 |

In Figure 3, the efficacy and behavior of channel capacity $C$ and EE in the massive MIMO system are demonstrated. In Figure 3a, the $C$ of resource allocation (RA) under the TCE model in [9], proposed SCE model, and the perfect CSI model are presented, respectively. The proposed SCE model and TCE model under different $\tau$ are also compared. From Figure 3a, the $C$ obtained by the proposed SCE model is higher than by the TCE model over the whole range of values shown. In the meantime, when PO $\tau = 0.1$, for $C = 60$ bit/s/Hz, the proposed SCE scheme surpasses TCE and perfect CSI scheme by 2.2 dB and –0.2 dB, respectively. The reasons are as follows: the proposed SCE scheme can directly eliminate the CSI distortion problem to source allocation and obtain the gain of CE in (40). As transmit power increases, the $C$ increases significantly, but the gains grow slowly. There is a smooth layer of $C$ with respect to $P$; hence, it is more meaningful to measure the $P_{\text{sum}}$ effectiveness by EE.

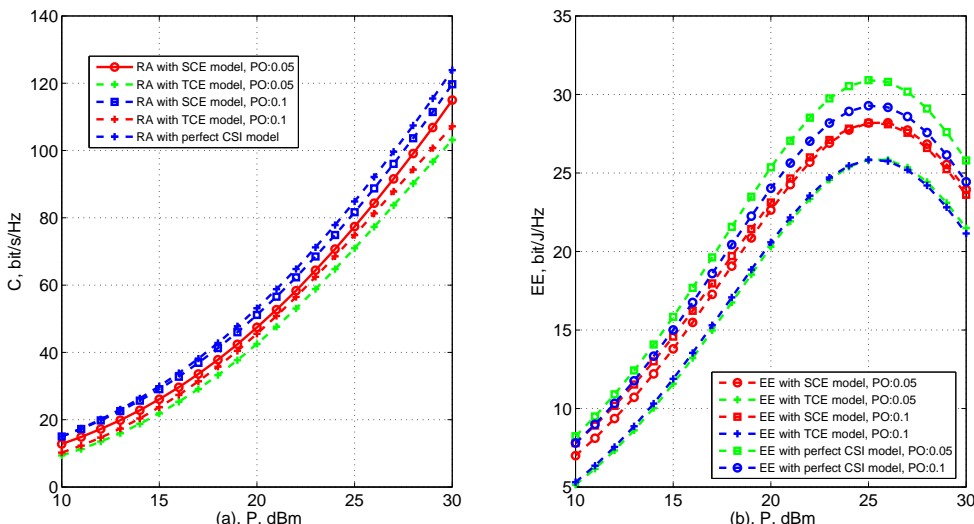

**Figure 3.** Comparison of the RA under the TCE model, SCE model, and perfect CSI model: (**a**) channel capacity versus different transmit power; (**b**) energy efficiency versus different transmit power.

In Figure 3b, the EE of RA under TCE, proposed SCE, and perfect CSI model are presented, respectively. As $P$ increases, the EE performance first increases and then decreases after reaching the maximum. It is obvious that the optimizing EE is a convex optimization problem and exiting the maximum EE boundary in valid coverage of $P$. By comparing the maximum boundary obtained by the proposed SCE scheme and TCE model when $\tau = 0.1$, the proposed SCE scheme can achieve the EE gain of 10%, and the optimal value of $P$ obtained by the proposed SCE scheme is in good agreement with the value obtained by the TCE scheme.

The WF threshold value in Equation (39) can be set to optimize the $C$, but the active antennas are redundancy and the EE decreases significantly. In Figure 4, the efficacy and behavior of the EE in Equation (41) based on antenna subset selection (AAS) and power allocation boundary (PAB) compared

with WF in the massive MIMO system are demonstrated. The maximum EE can be obtained by using the Quasi-Newton iteration method. It is observed that the EE optimization is a convex optimization problem and the maximum EE boundary is existing under the multidimensional rules condition. For a clearer comparison of the maximum EE of the proposed ASS and PAB based RA, WF based RA under the SCE model, the green curve and red curve are indicated, respectively, when $\tau = 0.05$. It is clear that the proposed scheme can outperform the WF based RA scheme, which shows the efficacy of the adjustive objective function. When the $C$ gain is small, the RF power consumption dominates EE, which can be chosen adaptively to guarantee EE. It can be clearly seen that the maximum EE boundary is different for different $P$, and proper $P$ can improve the EE effectively. Figures 5 and 6 show the proposed scheme and WF scheme under the TCE model and perfect CSI model with the same parameters. For clarity from Figure 4 to Figure 6, when $P$ are adaptively optimized, the EE performance by varying $|\psi|$ are presented in Figure 7. The EE performance first increases and then decreases after reaching the maximum as the number of antennas increases. For example, when the maximum EE is considered, the proposed scheme under the SCE model can achieve 5% EE gains compared to the TCE model, and 6% less than EE gains under the perfect CSI model.

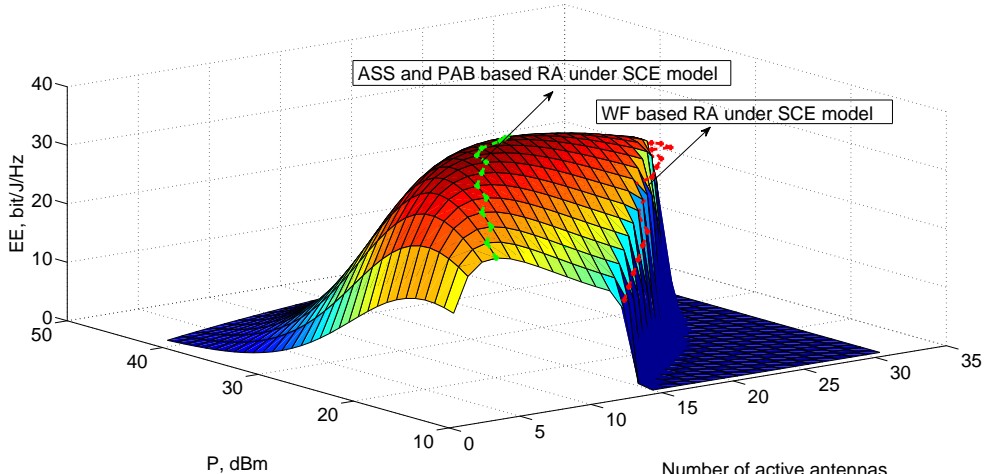

**Figure 4.** Energy efficiency with proposed ASS and PAB based RA under the SCE model for different number of activated antennas and transmit power.

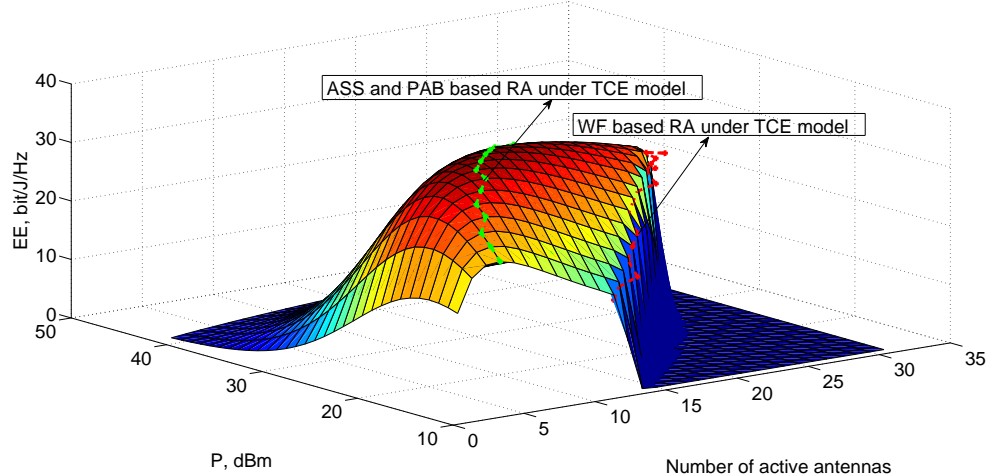

**Figure 5.** Energy efficiency with proposed ASS and PAB based RA under the TCE model for different number of activated antennas and transmit power.

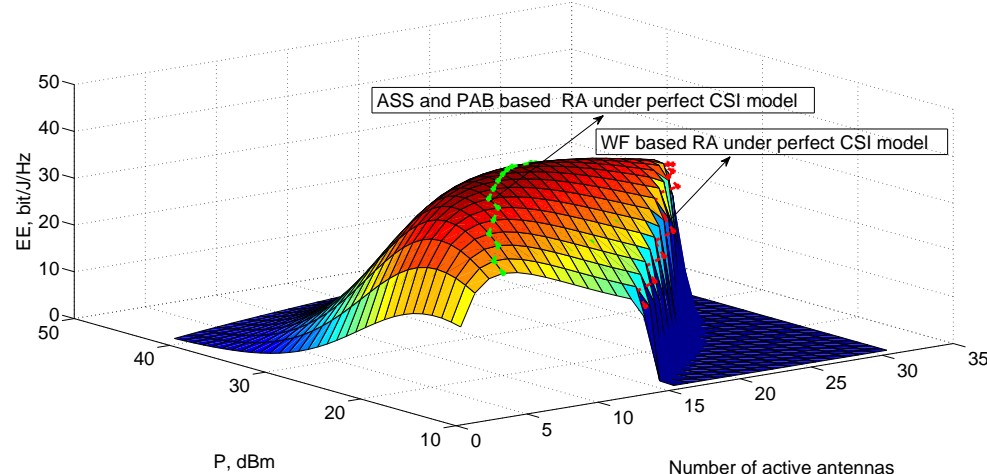

**Figure 6.** Energy efficiency with proposed ASS and PAB based RA under the perfect CSI model for different number of activated antennas and transmit power.

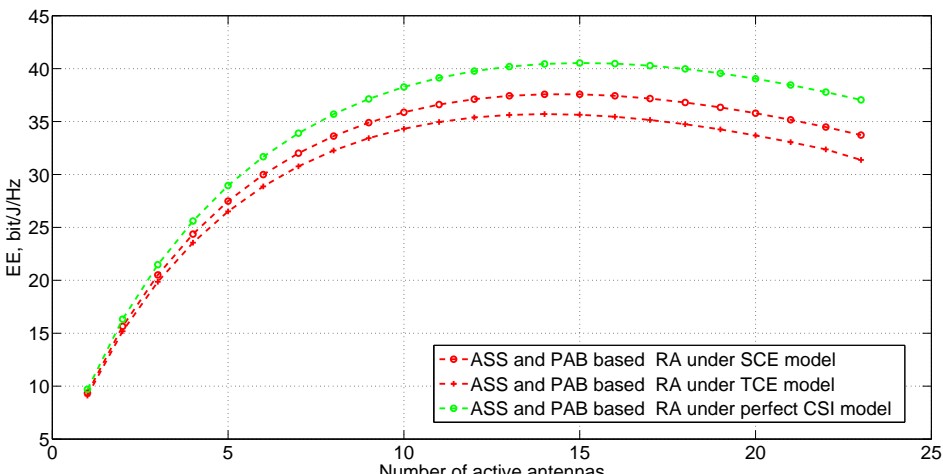

**Figure 7.** Energy efficiency with proposed SCE, TCE, and perfect CSI model for different number of activated antennas.

In Figure 8, the maximum EE of the proposed ASS and PAB based RA, WF based RA under TCE, proposed SCE, and the perfect CSI model in the massive MIMO system are demonstrated, respectively. When $|\psi|$ are adaptively optimized, the EE performance by varying $P$ is presented. It is observed that the maximum EE under the SCE model is closest to the ideal. From Figure 8, the maximum EE obtained through the proposed scheme surpasses the WF scheme under the SCE model by 23%. The simulation results show that the maximum EE obtained through the proposed RA strategy under the SCE model surpasses the strategy 5% when TCE is chosen and 6% less than the perfect CSI condition.

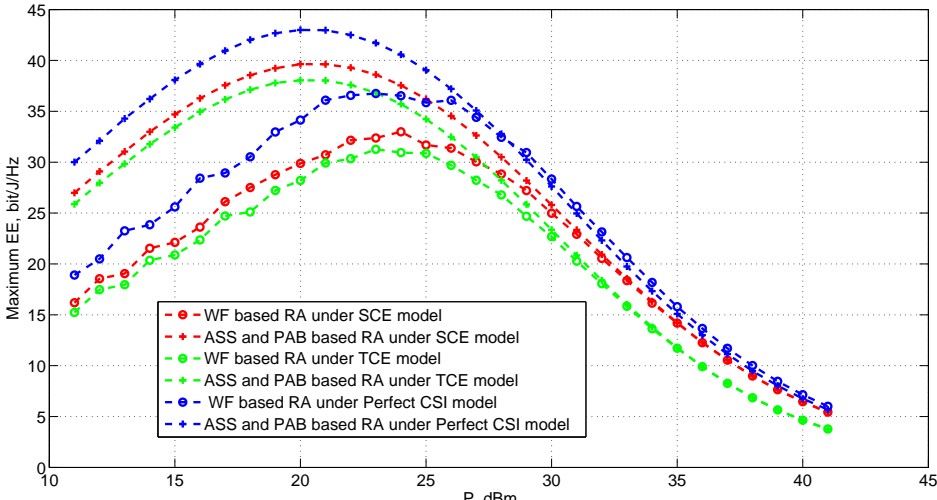

**Figure 8.** The maximum energy efficiency comparison for different schemes and models.

## 7. Conclusions

In this paper, the problem of resource allocation in two-way links Massive MIMO Systems with imperfect CSI is considered. The design of separated and bi-directional estimation can significantly increase the accuracy of CSI feedback and enhance the system capacity margin. Then, in order to maximize EE of a massive MIMO system by optimizing the optimal transmit power allocation, the pilot assignment and the number of available antennas have been investigated. Owing to the formulation of resource allocation in EE, the maximum EE obtained through the proposed ASS and PAB based RA scheme surpasses the WF based RA scheme under the SCE model by 23%. The simulation results show that the maximum EE obtained through the proposed RA strategy under the SCE model surpasses the strategy 5% when the TCE model is chosen and 6% less than the perfect CSI condition.

**Author Contributions:** F.H. and K.W. conceived and designed the experiments; K.W. performed the experiments; F.H. and S.L. analyzed the data; L.J. contributed the laboratory equipment; K.W. and F.H. wrote the paper. All authors have read and agreed to the published version of the manuscript.

**Funding:** This work was supported by "the Fundamental Research Funds for the Central Universities".

**Acknowledgments:** The authors would like to thank the Editor and the anonymous reviewers for their valuable comments and suggestions.

**Conflicts of Interest:** The authors declare no conflict of interest.

## Abbreviations

The following abbreviations are used in this manuscript:

| | |
|---|---|
| MIMO | Multiple Input Multiple Output |
| BS | Base Station |
| EE | Energy Efficiency |
| CC | Channel Capacity |
| CSI | Channel State Information |
| CE | Channel Estimation |
| FDD | Frequency Division Duplexing |
| TDD | Time Division Duplexing |
| WF | Water Filling |
| TCE | Traditional Channel Estimation |
| RA | Resource Allocation |
| SCE | Separated Channel Estimation |
| SVD | Singular Value Decomposition |

| EVD | Eigenvalue Decomposition |
|-----|--------------------------|
| SNR | Signal-to-Noise Ratio |
| SINR | Signal-to-Interference-plus-Noise Ratio |
| MSE | Mean Square Error |
| MMSE | Minimum Mean Square Error |
| ZF | Zero-forcing |
| OP | Orthogonal Procrustes |
| CRB | Cramer–Rao Bound |
| AAS | subset selection selection |
| PAB | power allocation boundary |

## Appendix A

The feasibility of AWGN model in uplink is described as follows:

Primarily, the transmitter can estimate the uplink CSI as $\hat{\mathbf{H}}_u$, based on the observation of uplink pilots. In addition, the CE is considered: $\mathbf{H}_u = \hat{\mathbf{H}}_u + \Delta\mathbf{H}_u$. As described above: $\hat{\mathbf{H}}_u$, $\mathbf{H}_u$, and $\Delta\mathbf{H}_u$ can be assumed as uncorrelated identically distributed (i.i.d.) complex white Gaussian with random variables.

The receiving signal in the uplink (BS) can be written as:

$$\begin{aligned}
\mathbf{y}_u &= (\hat{\mathbf{H}}_u + \Delta\mathbf{H}_u)\mathbf{x}_u + \mathbf{N}_u \\
&= \hat{\mathbf{H}}_u\mathbf{x}_u + \underbrace{\Delta\mathbf{H}_u\mathbf{x}_u + \mathbf{N}_u}_{\mathbf{W}_u}
\end{aligned} \tag{A1}$$

where $\mathbf{N}_u$ represents the AWGN noise in the uplink.

Using Equations (8)–(10) in Section 3.1, the equivalent observation $\mathbf{W}_u$ can be described as complex white Gaussian noise. Consider that $\mathbf{N}_u$, $\mathbf{H}_u$, and $\Delta\mathbf{H}_u$ are mutually independent and steady-state distribution in a certain time. In particular, $\mathbf{H}_u$ will not be adopted in EE optimization. While the equalization at the BS is designed by applying the well-known minimum mean squared error (MMSE) or zero-forcing (ZF) criterion to cope with the interference in the uplink, the effect of additive noise $\mathbf{N}_u$ and the CE error $\Delta\mathbf{H}_u$ can be approximated to AWGN:

$$\mathbf{y}_u = \mathbf{x}_u + \mathbf{z}_u \tag{A2}$$

where $\mathbf{z}_u$ is the uplink channel matric and satisfies: $\mathbf{z}_u \sim \mathcal{CN}(0,\sigma_{n_u}^2)$.

Considering that perfectly digital modulation and error-correcting mechanisms are being utilized in the uplink feedback, the AWGN $\mathbf{z}_u$ will be approached to 0. Assume that a flat channel constrains the time delay, when digital modulation and error-correcting mechanisms are utilized. For the assumption without error-correcting mechanisms, $\sigma_{n_u}^2$ can be fully exploited.

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
