# Peer review of "Energy Efficiency-Oriented Resource Allocation for Massive MIMO Systems with Separated Channel Estimation and Feedback"

_electronics, doi:10.3390/electronics9040582_

Round 1

Reviewer 1 Report

The paper by Hu et al. propose a dynamic resource allocation scheme to maximize the energy efficiency for Massive MIMO systems.

I recommend this paper for publication with minor revisions as follows:

1) Detail simulation model and condtions should be described for reader to clearly understand the results and reproduce them.

2) Clearly describe the related work and compare the representative related work with the proposed scheme.

3) I recommend that '8.Patents' should be deleted.

Author Response

Thank you very much!

Reviewer 2 Report

Summary:
The authors propose a new resource allocation scheme for energy efficiency in massive MIMO systems, with two parts: a new channel estimation (CE) scheme and a new power allocation scheme.
Regarding the CE scheme, the authors propose the receiver to transmit the received pilot back to the transmitter, which in turn will produce a channel state estimate H^.
In doing so, the sent-back pilot will be yet again affected by the channel, resulting in H_bar; the authors retrieve the actual channel estimate H^ at the receiver by mapping the problem into the orthogonal procrustes problem.

Comments:
Major:
1. My main concern is the claim that authors make in line 33, to motivates their whole CE solution: "There are non-ignorable CE errors under actual transmission conditions, which will significantly affect the SE/EE loss [8,9]".
For me, it is not clear what kind of errors are these.
    1.1 Going through the referenced papers doesn't provide relevant information about the source of these errors. Both papers use uplink pilots (client to BS) for the BS to estimate the downlink channel (i.e., the BS performs estimation implicitly), where in the current paper an explicit feedback scheme is used, i.e., transmitter sends pilot and a feedback is sent back. As far as I found, the referenced papers do not mention potential error sources for the explicit feedback, but the authors are welcome to point it out, and to some extent have sufficient information in their own paper for this to become clear.
   1.2 I'd assume, in most schemes with explicit feedback, the CS information is encoded digitally (and/or has some level of resilience to the channel). The model presented by the authors for the error affecting the uplink distortion for feedback (Section 3.2) seems to be very simplistic and failing to take into account modulation (and respective bit error ratio probability curves) and possible error-correcting mechanisms.
In sum, the authors should clarify in much more detail: how realistic is their error model affecting the feedback CSI (Eq. 14, in particular the model for z_u); how CSI is effectively transmitted in real-world system; how is Equation 15 produced.
   1.3 Sending the received pilot back to the transmitter, as the authors propose, seems a much more error-prone solution than sending the CE computed by the receiver. I understand the authors's option may facilitate bi-directional CE; however:
  (i) the subsequent resource allocation scheme only considers downlink estimates;
  (ii) the authors do not show evidence that their method performs better than existing bidirectional solutions (e.g., less overhead).

Again, I stress the authors to revisit their model for the errors affecting the CSI information sent back to the transmitter (not the sent-back pilot, but the processed CSI information itself), motivate it more deeply (in Section 3.2), and evaluate if their solution outperforms such a scheme.

2. It is not clear to be the advantage brought by the separate CE method for the power allocation mechanism. I guess some better tolerance to errors brought by the separate CE solution may improve the power allocation, but I've commented at length in the previous point my reservations about the proposed CE scheme.

3. Considerable revision of English should be made. I pointed out a few errors in the following 'Minor comments' section, but at some point I found so many that I stopped writing them down.

Minor:
- Line 98 (and in many other places): \tilde(z) \sim [something missing?] (0,\sigma_n^2) ; typically there's a letter in calligraphical font indicating the type of distribution (e.g., N for normal)

- Line 36: there seems to be a comma missing: "(...) uplink, utilizing (...)"
- Line 53: "(...) can be proven to be (...)"?
- Line 87: "WF" (water filling) -> acronym shows up here for the first time, but it is not written in full
- Line 136: "is existing" -> "exists"? (also in pg.14, line 236)
- Line 139: matric -> matrix (also in sentence between Eq.19 and Eq.20)
- Line 140: "(...) affect the CC loss." -> "(...) lead to decrease of CC."
- Line 146: "Differ from Figure 1 (...)"? Please revise English.
- Pg.7, first paragraph (for some reason lines are not numbered): "Precisely" -> "More concretely" ?
- Pg 10, first line of Sec. V: "The power efficiency decreases when the transmit antennas increases in [a] practical system." ?
- Pg.12, line 220: eliminating -> eliminate
- Pg.12, line 230: TCF?
- Pg.14, line 233: selection x2

PS: why are there not side numbers for all lines in the paper (e.g., between 96 and 97)? This was a bit upsetting. Not sure if authors' or template's fault.

Round 2

Reviewer 2 Report

Point 1.1 The authors point out that their scheme applies to FDD schemes.
The difference w.r.t. related work is clarified and clearly stated.

Point 1.2. The authors provide additional description of the error model for the feedback channel.
My remark was mostly about the sources of this error; by suggestion of the authors, I consulted the references from which this model was drawn (particularly ref. 16, previously 14) and clarified my doubts.
As a final comment, I suggest the authors clarify what is the concrete format of the feedback sent to the BS (e.g., ref. 16 describes it as "[unquantized] channel coefficients, transmitted as real and imaginary parts of a complex modulation symbol (...)").

Point 1.3. I appreciate the authors' reply and accept it.

Point 2. Thank you for the clarification that channel reciprocity may not hold in FDD channels.
